# Critical Assessment of Mycotoxins in Beverages and Their Control Measures

**DOI:** 10.3390/toxins13050323

**Published:** 2021-04-30

**Authors:** Md. Shofiul Azam, Shafi Ahmed, Md. Nahidul Islam, Pulak Maitra, Md. Mahmudul Islam, Dianzhen Yu

**Affiliations:** 1SIBS-UGENT-SJTU Joint Laboratory of Mycotoxin Research, CAS Key Laboratory of Nutrition, Metabolism and Food Safety, Shanghai Institute of Nutrition and Health, University of Chinese Academy of Sciences, Chinese Academy of Sciences, Shanghai 200031, China; shofiul.sust@gmail.com; 2Department of Agro Product Processing Technology, Jashore University of Science and Technology, Jashore 7408, Bangladesh; shafi@just.edu.bd; 3Department of Agro-Processing, Bangabandhu Sheikh Mujibur Rahman Agricultural University, Gazipur 1706, Bangladesh; nahidul.islam@bsmrau.edu.bd; 4Solid-State Fermentation Resource Utilization Key Laboratory of Sichuan Province, Faculty of Agriculture, Forestry and Food Engineering, Yibin University, Yibin 644000, China; pulak@yibinu.edu.cn; 5State Key Laboratory of Respiratory Disease, Guangzhou Institutes of Biomedicine and Health, Chinese Academy of Sciences, Guangzhou 510530, China; mislamru@gmail.com

**Keywords:** contamination, aflatoxins, ochratoxin A, patulin, toxicity, detoxification

## Abstract

Mycotoxins are secondary metabolites of filamentous fungi that contaminate food products such as fruits, vegetables, cereals, beverages, and other agricultural commodities. Their occurrence in the food chain, especially in beverages, can pose a serious risk to human health, due to their toxicity, even at low concentrations. Mycotoxins, such as aflatoxins (AFs), ochratoxin A (OTA), patulin (PAT), fumonisins (FBs), trichothecenes (TCs), zearalenone (ZEN), and the alternaria toxins including alternariol, altenuene, and alternariol methyl ether have largely been identified in fruits and their derived products, such as beverages and drinks. The presence of mycotoxins in beverages is of high concern in some cases due to their levels being higher than the limits set by regulations. This review aims to summarize the toxicity of the major mycotoxins that occur in beverages, the methods available for their detection and quantification, and the strategies for their control. In addition, some novel techniques for controlling mycotoxins in the postharvest stage are highlighted.

## 1. Introduction

Mycotoxins are naturally occurring, poisonous compounds produced from filamentous fungi or molds that can be found in foods. Mycotoxins have a huge set of chemical compounds generated by diverse mycotoxigenic fungi species [1]. Over 400 toxic metabolites are produced by more than 100 fungi species [2]. Humans are exposed to mycotoxins through the consumption of contaminated foods [3]. They can pose negative health effects, ranging from acute toxicity to chronic symptoms, such as kidney damage, liver damage, immune deficiency, and cancer [4,5]. The mostly commonly identified mycotoxins in fruit juices and beverages are Aflatoxins (AFs), Ochratoxin A (OTA), Patulin (PAT), Fumonisin (FB), Trichothecenes (TCs), Zearalenone (ZEN), and the Alternaria toxins, which are mainly produced by *Aspergillus* (AFs, OTA, PAT), *Penicillium* (OTA and PAT), *Fusarium* (DON, ZEN, FB), *Byssochlamys* (PAT), and *Alternaria* species. Most of the mycotoxins are immunosuppressive substances that are categorized as neurotoxins, nephrotoxins, hepatotoxins, or carcinogens. Several factors, such as the species, strain of fungi, matrix composition, temperature, and moisture content determine the characteristics and functionalities of the toxins [6].

Cereal grains and fruits can be infected by molds at various stages of production, for example, during cultivation, harvesting, and storage [7]. The contamination of mycotoxins is a worldwide problem, but it is more serious in humid and warm environmental conditions that favor the growth of fungi and production of mycotoxins. In a recent study, 60% to 80% of agricultural products were found to be contaminated by mycotoxins [8]. Moreover, nearly one-fourth of agricultural food products worldwide are contaminated by mycotoxins at a concentration above the Codex Alimentarius and EU maximum levels [2]. As secondary metabolites, mycotoxins are very durable chemical components that can be transmitted from raw materials to processed products such as beverages, which can pose a serious health risk to consumers (Figure 1).

Over the last few years, distinguishable progress in society has driven reforms in the world beverage market. Consumers are becoming more conscious about the effect of diet on their health. Beverages are not only responsible for providing energy and hydration but also for strengthening health and preventing nutrition-related defeciencies [9]. The application of effective measures to protect consumers from the toxic effects of mycotoxins and, subsequently, to defend against public health is very significant and crucial. In the last few years, intensive research has been conducted to explore different techniques for the prevention of fungal infection and mycotoxin contamination in food. Fungal infection can be prevented at an early stage of production and during postharvest storage of crops. Detoxification of food through different processing techniques must be considered if fungal infection cannot be avoided, as it is the final defense mechanism for the prevention and detoxification of mycotoxins in foods for human consumption. Therefore, this comprehensive review mainly focuses on (a) the adverse health effects of some important mycotoxins associated with beverages; (b) different sophisticated methods for mycotoxin detection and quantification; (c) different potential strategies including biological, chemical, and physical strategies and their combination for the prevention and detoxification of mycotoxins in beverages. Some state-of-the-art approaches that can be applied in food processing including biocontrol, enzymatic control, and adsorption by biological materials are also discussed.

## 2. Major Mycotoxins in Beverages

### 2.1. Aflatoxins

Aflatoxins (AFs) are mainly produced by *Aspergillus* spp. In most of the cases, contamination with AFs takes place after harvesting and during storage. Inappropriate management during transportation and storage including exposure to conditions such as high humidity (˃65%) and temperatures rapidly increases the AF concentration in food. The four major naturally occurring aflatoxins are AFB1 (classified as Group 1 carcinogen by the IARC), AFB2, AFG1, and AFG2 [6]. The disease caused by the consumption of AFx-contaminated food is generally known as aflatoxicosis, and the acute symptoms in humans include abdominal pain, vomiting, pulmonary, icterus, coma, cerebral edema, convulsions, hemorrhage, and even death. In the case of chronic aflatoxicosis, it suppresses the immune system and induces tumors or other adverse pathological conditions [10]. Moreover, continuous exposure to AFs is significantly correlated with cognitive impairment and growth inhibition in children [10]. Hence, the U.S. Food and Drug Administration (FDA) rigidly regulates the incidence of AFs in human food, and the AF level in foods should be less than 20 ppb [11]. The European Food Safety Authority (EFSA) has set a stricter tolerance level for AFs (maximum of 2 μg/kg for B1 and 4 μg/kg for total AFs) in processed food [12].

### 2.2. Ochratoxin A

Ochratoxins (OTs) are group of mycotoxins that are mostly generated by *Aspergillus* and *Penicillium* species. OTs are classified into three groups, Ochratoxin A, B, and C, based on the characteristic functional group. Ochratoxin A (OTA) is the most toxic member of the OT family [10]. OTA is a vital nephrotoxic mycotoxin with genotoxic, hepatotoxic, immunotoxic, teratogenic, carcinogenic, and possibly neurotoxic effects. OTA can inhibit the synthesis of some proteins as well as Deoxyribonucleic acid (DNA) and Ribonucleic acid (RNA) [10]. The IARC classified OTA as Group 2B, possibly carcinogenic to humans. The EFSA has set the maximum levels of OTA at 2 mg/kg for beverages including wine and grape juice for direct human consumption [13]. Good agricultural, harvesting, and storage practices, especially avoiding physical and physiological damage, can prevent the incidence of OTA in fruit juices [14].

The occurrence of OTA-producing fungi and the level of OTA may vary with the climatic conditions [15]. OTA is generally found in subtropical areas and in high-temperate climate regions and can be available in various food products in these areas, for example, beer, wine, and grape products [14]. It has been noticed that in the subtropical regions of Argentina, Australia, and Brazil, OTA is triggered by the black *Aspergilli*, *Aspergillus carbonarius* [16]. As with other mycotoxins, OTA is comparatively stable within the range of heat treatments in the food processing industry [17]. The incidence of OTA has been reported in white, rose, and red wines obtained from the Muscat Italia, Syrah, and Touriga Nacional varieties, respectively [18]. Table 1 summarizes the major mycotoxins responsible for the contamination of beverages. 

### 2.3. Patulin

Patulin (PAT) is predominantly generated from various *Penicillium*, *Aspergillus*, and *Byssochlamys* species and possesses various hazardous features such as toxicity, carcinogenicity, and mutagenicity. *P. expansum*, *B. fulva*, and *B. nivea* are significant PAT-producing microorganisms. Patulin has been identified in many foods, particularly in fruits and beverages [34]. The higher amounts of moisture and sugar in fruits stimulate the generation of PAT [23,35,36,37]. It is highly soluble in water and very stable in aqueous acidic media, so it is basically found in apple-derived products such as apple juices. Since this mycotoxin is resistant to heat, pasteurized fruit juices may also contain PAT as a secondary metabolite of *Byssochlamys* spp. [38]. Washing of fruits and discarding of rotten portions are two primary low-cost practices used by the industries to eliminate the incidence of PAT in manufactured fruit products such as fruit juices and concentrates, jams, and apple-based drinks [39]. PAT is degraded by the fermentation reaction during cider or wine manufacturing processes [40]. PAT is associated with subacute and acute toxicity and chronic symptoms. Symptoms of acute toxicity include dyspnea, anxiety, edema, distension and hyperemia of the gastrointestinal tract, pulmonary congestion, intestinal inflammation, intestinal hemorrhage, and epithelial cell degeneration [41]. Chronic toxicity of PAT includes immunotoxicity, genotoxicity, hepatotoxicity, steroid toxicity, dermal toxicity, enterotoxicity, neurotoxicity, oncogenicity, nephrotoxicity, teratogenicity, and embryotoxicity [36,42]. Moreover, PAT can create lesions in many body tissues/organs, including the liver, kidneys, heart, lungs, brain, embryos, ovaries, skin, bones, and thyroid. Therefore, controlling the incidence of PAT in food and preventing consumers from exposure to PAT is very important.

### 2.4. Fumonisin

Fumonisin (FB) mycotoxins are secondary metabolites of *Fusarium* spp, mostly *Fusarium verticillioides* and *F. proliferatum.* The group of 30 homologs of FBs is divided into four main groups: fumonisin A, B, C, and P [24,43]. The fumonisin B group occurs in nature with the highest frequency and includes fumonisin B1 (FB1), B2 (FB2), and B3 (FB3) [44]. FB1 is the most toxic member of the FB family [43]. Fumonisins are cytotoxic and carcinogenic to humans and animals. FB1 has been associated with esophageal cancer in different countries [45,46,47]. It was found to be a contaminant of wheat, corn, and barley. A recent study conducted by Piacentini et al. [48] quantified a very high concentration of FB1 in beer (four times higher than the maximum allowable concentration). According to the IARC, FB1 is possibly carcinogenic (Group 2B) to humans [6]. The US FDA has established the maximum advisory level of 2 to 4 mg/kg for FB.

### 2.5. Trichothecenes

Trichothecenes (TCs) belong to a large group of structurally related toxins, mainly produced by fungal species of the *Fusarium* genus [49]. TCs, due to their amphipathic nature and low molecular weights, are easily absorbed by the intestinal membranes and quickly dispersed to different parts and tissues of the body [50]. More than 200 different TCs are classified into four different groups: the richothecenes A, B, C, and D. Among trichothecenes, types A and B occur most frequently and are the most harmful mycotoxins that contaminate foods. Type A trichothecenes (T-2 and HT-2) are the most toxic group of TCs [51].

#### 2.5.1. Type A Trichothecenes (T-2 Toxin and HT-2)

T-2 and HT-2 toxins are mainly produced by *Fusarium langsethiae*. *F. sporotrichioides*, *F. poae*, and *F. acumninatum*, and *F. equiseti* have also been observed to generate T-2 and HT-2 toxins [52,53]. T-2 and HT-2 toxins have been detected in barley, oat, maize, wheat, rice, beer, and plant-based milks, especially in oat- and soy-based milks and beverages [28,49,54,55]. T-2 and HT-2 toxins are linked to hematotoxicity, myelotoxicity, and growth retardation [56]. The EFSA established a TDI value of 100 ng/kg BW/day for HT-2 and T-2 toxins [53].

#### 2.5.2. Type B Trichothecenes (Deoxynivalenol)

Deoxynivalenol (DON) is synthesized by different species of the *Fusarium* genus, mainly by *Fusarium culmorum* and *Fusarium graminearum* in cereals [57]. It also contaminates cereal-based food products, for instance, pasta, bread, and beer. Acute gastrointestinal symptoms due to the consumption of foods contaminated with DON include headache, nausea, dizziness, vomiting, abdominal pain, hemorrhagic diarrhea, and fever [58]. Long-term dietary exposure to DON also causes different toxicological effects in humans and animals such as hepatotoxicity, anorexia, dermatological problems, and ribotoxic stress [59,60]. DON is the most frequent and represents the highest public health concern related to the consumption of beer [27,61]. The US FDA has established the maximum advisory level of DON as being 1 mg/kg. The EFSA recently established a provisionally maximum tolerable daily intake (PMTDI) of 1 µg/kg BW/day for DON [62].

### 2.6. Zearalenone

Zearalenone (ZEN) is produced by various species of *Fusarium*, mainly *F. graminearum*, *F. culmorum*, and *F. sporotrichioides*. It infects corn, wheat, barley, oat and rye, mainly in areas with temperate climates [63]. ZEN is a nonsteroidal, estrogenic mycotoxin that affects ovulation, conception, and fetal development at concentrations above 1 mg/kg [64]. ZEN causes hyperestrogenism, mainly affecting reproduction, and some studies have linked ZEN with the stimulation of human breast cancer cell growth [65]. Mycotoxins can be transferred from cereals to malt and then to beer due to their water solubility (FB and DON) and high thermal stability (AFs, DON, and ZEN) [66]. The EFSA established a TDI for ZEN of 0.25 µg/kg BW/day [50]. Under EU legislation, the ZEN concentration should be less than 20 to 200 μg/kg, depending on the food category [67].

### 2.7. Alternaria Toxins

The main Alternaria mycotoxins are Tenuazonic acid (TeA), Alternariol (AOH), and alternariol monomethyl ether (AME). *Alternaria* spp., mainly *Alternaria alternata*, *A. tenuissima*, and *A. arborescens* produce Alternariols and are found in a large range of foods including berries, prune nectar, carrot juice, apple juice concentrate, grape juice, raspberry juice, cranberry juice, beer, and red wine [68,69]. Alternaria mycotoxins are toxic, mutagenic, and carcinogenic and are responsible for DNA helix deformation, sphingolipid metabolism disorder, inhibition of protein synthesis, and photophosphorylation [70,71].

## 3. International Standards for Mycotoxins in Fruit Juices and Beverages

The joint Scientific Expert Committee on Food Additives of the World Health Organization (WHO) and the Food and Agriculture Organization (FAO) of the United Nations—called the JECFA—is the international body responsible for health risk assessment from mycotoxins. The Codex Alimentarius Commission has established the international standards to limit exposure to mycotoxins from certain foods based on JECFA assessments. The JECFA set a provisional maximum tolerable daily intake (PMTDI) for PAT of 0.4 μg/kg BW/day [72,73]. The maximum limit for PAT set by Codex for apple juice is 50 µg/L. The food and drug administration (FDA) in the USA has recommended that apple juice, apple juice concentrate, and apple products should not contain a residual PAT concentration of more than 50 μg/L or 50 ppb. Apple juice products containing more than 50 ppb for adults or 10 ppb for infants and young children are considered adulterated and are designated as harmful for adult human health [74]. The maximum limit for total AFs set by the US FDA is 20 µg/kg [30]. The maximum limits for AFs according to Codex in various grains, dried figs, and milk range from 0.5 to 15 µg/kg. As a consequence of the presence of mycotoxins in various foods and their human health risk, the maximum limits for mycotoxins in foodstuffs have also been determined by the EU (Table 2).

## 4. Detection and Quantification of Mycotoxins in Beverages

In most cases, mycotoxin levels in contaminated food and beverages can be very low, and this necessitates the development of a suitable, rapid, and sensitive detection method. Various analytical testing procedures have been developed for mycotoxin detection and quantification due to their diverse forms [75]. Normal chromatographic procedures are usually time consuming and cost intensive; therefore, a range of methods, mostly based on immunological principles, have been developed and commercialized for quick determination [76]. Some common mycotoxin detection methods in beverages as well as beverage-producing crops are summarized in Table 3. 

### 4.1. HPLC Detection

A technical expert team from the Shimadzu Corporation developed an automated screening system from pretreatment to analysis and final reporting of the analytical results for the simultaneous analysis of 10 commonly tested mycotoxins, AFB1, AFB2, AFG1, AFG2, AFM1, OTA, PAT, ZEN, deoxynivalenol (DON), and nivalenol (NIV), from grape juice, apple juice, and beer with an HPLC system in 14 min. They mainly used the HPLC system for mycotoxin detection and separation from 10 different beverage samples. Fluorescence (FL) and photodiode array (PDA) are the main detection techniques involved in this analysis system [99]. Another recent study combined different steps of mycotoxin analysis, such as centrifugation and shaking of liquid and solid dispensing units into a single automatic and robotic platform. Automated analysis and detection systems have several benefits, such as cost effectiveness, time saving, better quality assurance, traceability, accuracy, and high efficiency [100].

### 4.2. Mass Spectrometry Methods for Beer Mycotoxin Detection

An analytical method was developed for the detection and quantification of 15 mycotoxins (AFB1, AFB2, AFG1, AFG2, AFM1, T-2 toxin, HT-2 toxin, OTA, PAT, NIV, DON, ZEN, FB1, FB2, and FB3) from beer-based drinks such as beer, low-malt beer, and new genres. First of all, mycotoxins were extracted from samples using acetonitrile with sodium chloride, anhydrous magnesium sulfate, and sodium citrate, and then purified with a solid-phase extraction (SPE) cartridge including HC-C18 [101]. Finally, mycotoxins were identified and quantified by a modified QuEChERS method and ultra-high-performance liquid chromatography combined with tandem mass spectrometry (UHPLC/MS/MS) [102]. Fourteen mycotoxins (AFB1, AFB2, AFG1, AFG2, AFM1, T-2 toxin, HT-2 toxin, OTA, PAT, DON, ZEN, and FB1, FB2, FB3) were identified and quantified in wines using two solid-phase extractions and ultra-high-performance liquid chromatography coupled with tandem mass spectrometry (UHPLC-MS/MS) within 13 min [103]. Miró-Abella et al. [28] documented a procedure for the simultaneous identification of 11 mycotoxins in plant-based beverage matrices such as soy, oat, and rice using QuEChERS extraction followed by ultra-high performance liquid chromatography coupled with tandem mass spectrometry (UHPLC–(ESI)MS/MS).

### 4.3. Biosensor and Immunosensor Mycotoxin Detection

The biosensor has some unquestionable advantages over conventional methods used for the detection of mycotoxins in foods, such as the quick detection of mycotoxins with no or minimal enrichment. They are easy to use and do not require highly skilled operators [104]. Joshi et al. [105] proposed a competitive inhibition immunoassay that can be used in the agricultural field or at-line for the detection of DON mycotoxins in beer and also in barley without preconcentration, while the detection of OTA in beer requires an extra enrichment step, which makes it less suitable for the detection of OTA in its present form. Pennacchio et al. [106] developed an efficient and cost-effective immunoassay method to measure PAT by connecting the immunological detection of PAT with an optical method known as surface plasmon resonance (SPR). The detection limit of this test in apple juice was 1.54 × 10^−2^ µg/L. However, a promising fluorescence polarization approach was developed by Pennacchio et al. [107], where near infrared (NIR) fluorescence sensors were used to identify PAT without the pretreatment of apple juice. A conductometric urease-based biosensor to track PAT inhibitory activity was developed by Soldatkin et al. [108] This biosensor is appropriate for determining PAT levels above 50 µg/L in apple juices, and has been reported to have comparatively high PAT sensitivity, quick signal reproducibility, and high selectivity.

A quick and sensitive immunochemical method was developed to identify the possibility of a transfer of tremorgenic paxilline (PAX), an indole-diterpene alkaloid mycotoxin, into beer from barley and rye. A competitive indirect enzyme immunoassay (EIA) was used to identify PAX in beer. The immunoassay is based on the specific antibodies for specific mycotoxin compounds in beverage samples [109].

### 4.4. Microchip Method for the Detection of Mycotoxins in Beverages

Man et al. [110] reviewed the use of microchips for mycotoxin determination, mainly involving optical, electrochemical, photo-electrochemical, and label-free recognition systems. The unique advantage of this method is the need for low amounts of sample and the low detection limit. Some microchip methods have limitations due to detection only being possible in the laboratory. Lab-on-a-chip is a suitable, accurate, and sensitive method for the detection of mycotoxin-infected samples [111]. This microchip is constructed from disposable microfluidic polymer chips and microarray chips that are mainly produced from inorganic (glass, silicon) and elastomeric polymer (PDMS) materials [112,113,114].

### 4.5. Biomarker Assay

The biomarker assay for mycotoxin detection in beverage products is mainly conducted after the exposure of humans to mycotoxins through consumption. Human urine and blood samples are mainly used for biomarker assays of mycotoxins [115]. High-resolution mass spectrometry, in addition to new analytical approaches, is the main step involved in biomarker assays, and it is mainly known as a biomonitoring assay. Isotopes can be used to assist biomarker assays when assessing multiple urinary samples [116,117]. The major mycotoxins (AFB1, OTA, DON, ZEN) that can infect fruits, vegetables, and their final products beverages can be identified with a novel microRNA (miRNA) biomarker assay. The regulatory roles of miRNAs in mycotoxin-induced toxicity have been studied using various novel biomarkers. This study helps us to understand the molecular mechanisms involved in apoptosis during the cell cytotoxicity assay [118].

### 4.6. Nanoparticle-Based Detection Methods

There are various kinds of nanomaterials used for the purification and detection of mycotoxins, such as gold nanoparticles, magnetic nanoparticles, nano-silver, carbon-based nanomaterials, metal oxide nanoparticles, and Quantum Dots (QDs) [119]. Gold and silver nanoparticles help to improve ELISA and aptamer-linked mycotoxin detection. AFB1 detection firstly produces an antibody–enzyme composite followed by an electrochemical system [75]. Nanomaterials are used for biosensors due to their small particle sizes and unique chemical, physical, and electronic properties [120].

### 4.7. ELISA Detection System

Conventional chromatographic techniques are usually expensive and time consuming, which necessitates the development of rapid detection techniques. Immunological methods, such as enzyme-linked immunosorbent assay (ELISA), have become the most powerful and useful analytical methods for the detection of mycotoxins due to their fast and economical measurements [98]. The ELISA method is based on the detection of a three-dimensional structure of a specific mycotoxin by a specific antibody [121]. Direct and indirect ELISA methods have been developed and commercialized for the detection of OTA, PAT, and fumonisin in wine and beer samples and also for aflatoxins and *Fusarium* toxins in cereals [122,123,124]. These test kits are accurate and trustworthy. One extracted sample can be used to detect six mycotoxins. A conventional ELISA microtiter plate needs the antibody–antigen reaction to be in equilibrium, which requires an incubation period of almost 1–2 h; at present, most of the commercially available ELISA test kits for mycotoxins operate in the kinetics phase of antibody–antigen binding, which shortens the incubation period to minutes. Their main advantages are that they are simple, cheap, have high sensitivity, allow the simultaneous analysis of multiple samples, and are easy to screen. The main disadvantage is that they require 30 samples to be tested per ELISA set. The commercially available ELISA kits are the AFM1 ELISA Kit and the I’screen AFM1 milk ELISA Kit [125,126].

## 5. Mitigation Policies of Mycotoxin Contamination in Beverages

Nearly all mycotoxins are thermally resistant and cannot be simply degraded by normal heat treatment methods during food processing or household cooking methods [26]. Normally, mycotoxin contamination in beverages can be controlled by preventing the contamination of agricultural raw materials used for the production of beverages [127,128]. Implementation of good manufacturing practices will help to ensure safe beverage production without mycotoxin residues. Good manufacturing practices (GMPs) include the use of proper sorting, processing, drying, cooling, and storage conditions for agricultural raw materials. Complete reduction in the number of mycotoxins, or at least a number not higher than the maximum allowable limits, can be achieved by different pre- and postharvest treatments [129]. It has been observed that some pathogenic bacteria, viruses, and parasites can survive in fresh agricultural products (fruits and berries), and juices manufactured from them and can create disorders after consumption. Hence, the prevention of growth of mycotoxin-generating molds and the detoxification of foods from mycotoxins by postharvest treatment have become important issues in food safety research. Postharvest contamination could be prevented by monitoring the temperature, moisture, humidity, microbial growth, and insect and pest infestation during storage. Various biological, physical, and chemical methods have been developed for the prevention or detoxification of mycotoxins during the postharvest period (Figure 2), although some techniques are less efficient and sometimes restricted due to safety concerns, possible degradation in the nutritional value of the treated products, and the cost of application [26,130].

In recent years, consumers have increasingly preferred tp consume organic foods. The trend for the consumption of fresh food products is growing day by day [131,132]. Therefore, it is very important to control the quality of food products during the postharvest period, as there is a high possibility of infection by mycotoxins that generate molds and their toxins during this period [41,133].

### 5.1. Physical Control Methods

Some novel physical approaches have been developed to reduce the mycotoxin concentration in beverages, such as irradiation, high pressure processing, and the use of adsorbents [23,39,134].

#### 5.1.1. Irradiation

Food irradiation not only prevents the growth of fungi but also degrades some mycotoxins [135,136]. Ionizing radiation can prevent the growth of fungi by altering their physiological functions or cell structures, for example, through the breakdown of DNA and mechanical damage of cell walls by destabilizing the lipid bilayer and proteins of cell membranes [137]. The efficacy of irradiation is controlled by different factors, such as the dose of irradiation given and the physiological stage and morphological structures of fungi. In a previous study, fruits and vegetables were inoculated with fungi, and then different doses of irradiation were applied to prevent fungal contamination. The results indicated that higher doses of irradiation led to better fungal inhibition in oranges and papayas [138,139]. However, product quality should be taken into consideration, because fruits have a tendency to lose their color and texture as a result of irradiation treatment. The application of UV radiation is very effective to reduce the PAT concentration in beverages. In an investigation on PAT detoxification using UVC wavelengths ranging from 200 to 280 nm in apple cider or apple juice, Zhu, et al. [140] reported a 90% reduction in the toxin. UV exposure also reduced ascorbic acid in apple juice by 36.5%. Assatarakul, et al. [141] measured the initial concentration of PAT (1000 ppb) in apple juice, and UV exposure at 14.2 mJ/cm^2^ and 99.4 mJ/cm^2^ resulted in decreases in the PAT level of 5.14% and 72.57%, respectively. UV irradiation considerably changed the total soluble solid (TSS) concentration, pH, and sensory quality, but the changes in ascorbic acid and the titratable acidity of apple juice were not significant. Kim, et al. [142] found that UV treatment with wavelengths between 200 and 280 nm for 5, 10, and 30 min reduced the PAT level in apple juice from an initial concentration of 94.11 𝜇g/L to 69.28, 54.55, and 5.92 𝜇g/L, respectively, and the PAT concentration in apple juice was reduced to below the detection level after 30 min of UV treatment.

Sensory properties, antioxidant properties, flavonoid and phenolic contents, ascorbic acid, and titratable acidity were taken into consideration to assess the fruit juice quality. The quality of every fruit juice parameter reduced slightly as dose of irradiation increased from 2.5 to 5 and then to 7.5 kGy, and the quality significantly deteriorated at an irradiation dose of 10 kGy. However, irradiation within a certain dose range might be applied for the degradation of mycotoxins in fruit juices [143]. In addition, the safety uncertainty due to the production of free radicals in treated products is not negligible, and it is also necessary to assess the toxicity of mycotoxin-degraded products. For instance, UVC (200 to 280 nm) treatment for an extended period could produce furans, which are categorized as possible carcinogens for humans [144]. Nevertheless, irradiation seems a promising method for reducing the mycotoxin level in fruit juices.

#### 5.1.2. Thermal Treatment

Most mycotoxins are heat resistant. Partial degradation of aflatoxins and OTA takes place at temperatures of around 250 and 200 °C, respectively. Fumonisins could be completely destroyed at temperatures of over 180 °C [145], and the degradation of DON occurs at 210 °C [146]. Generally, the degradation of mycotoxins depends on the duration and temperature of treatment. Thermal treatment can be combined with high-pressure processing (HPP) to accelerate the degradation of mycotoxins in foods.

#### 5.1.3. High-Pressure Processing

The contamination of food with heat-resistant fungi (HRF) and their spores, commonly found in fruit beverages and concentrates, is a major concern for fruit processing and food safety issues, leading to significant economic losses. HPP is an emerging nonthermal food-processing technology that acts as an alternative to conventional thermal processing techniques to meet consumers’ demands for minimally processed and fresh-like food products. HPP retains the freshness, flavor, texture, appearance, and color of foods and reduces the loss of nutrients when compared with thermal processes, as no heat is applied to foods during HPP treatment [147]. Nowadays, HPP is widely used to process fruit juices and beverages as a nonthermal food pasteurization procedure. The US FDA has approved HPP for use as a non-thermal pasteurization method. During the commercial application of HPP, the pressure levels vary from 100 to 1000 MPa and the method can work at temperatures between −20 and 90 °C. HPP typicaly degrades mycotoxins by altering their chemical structures. HPP treatment (600 MPa at 11 °C for 300 s) was shown to reduce PAT by 10.9%–25.5% in fruit juice blends [148]. The treatment of apple juice with HPP (300–500 MPa for 5 min at 20–50 °C) has been reported to reduce the PAT concentration by up to 51% [149].

#### 5.1.4. Pulsed Light Technology

Pulsed light (PL) is an emerging nonthermal food processing/preservation technology that involves the discharge of short and high-intensity pulses of light into the food product for the detoxification of mycotoxins. PL minimize the deleterious effects of thermal processing on food quality while preserving the nutritional and sensorial attributes of food [150]. The detoxification effects of PL on mycotoxins are attributed to the fragmentation of mycotoxin molecules [151,152,153]. Apple juice and apple purée treated with PL doses of 2.4 and 35.8 J/cm^2^ resulted in up to reductions in PAT of 22% and 51%, respectively [154].

### 5.2. Chemical Control Methods

#### 5.2.1. Ozone Treatment

Ozone (O_3_) is a strong oxidizing agent that is usually considered a safe antimicrobial agent in the food and beverage industries [155]. The major mycotoxins, including AFs, OTA, PAT, FB1, ZEN, and DON, which are stable at conventional food processing treatments, can be destroyed by ozone within minutes,. Ozone changes the molecular structures of mycotoxins and forms products with less toxicity. OTA, PAT, and ZEN degraded byproducts of ozone treatment could not be detected by fluorescence or UV detection. The efficacy is not only dependent on the ozone concentration and duration of exposure but also on the properties of food products, the temperature, the moisture content, the pH, and the relative humidity [156]. Some benefits of ozone above any other chemical oxidizing agents are as follows: (a) both gaseous and aqueous forms of ozone are available for application, (b) there are many ozone precursors, (c) ozone treatment does not produce any residue, (d) there is no associated hazardous disposal, and (e) ozone can be produced onsite [157,158]. Diao, et al. [159] applied ozone for the degradation of PAT in apple juice and observed that exposure to 7 and 12 mg/L of O_3_ gas for 10 min reduced the PAT concentration in apple juice by 64.77% and 81.66%, respectively. Detoxification of mycotoxins by ozone was shown to decrease the total phenolic content, the malic acid and ascorbic acid concentrations, and the color of apple juice, but the pH, total acid content, and soluble solid content did not change significantly. However, ozone treatment can be applied as a safe and green technology to control mycotoxins [155].

#### 5.2.2. Use of Chemical Adsorbents

Some chemicals adsorb mycotoxins due to their weak interactions with mycotoxins. Chemical adsorbents such as propylthiol-functionalized SBA-15 silica [160], magnetic carbon nanotubes (Fe_3_O_4_− MWCNTs adsorbent) [161], and sulfhydryl-terminated magnetic bead separation [162] are more often used. Hydrated sodium calcium aluminosilicates (HSCASs), obtained from natural aeolite are the most commonly used clay-based adsorbents [163]. These adsorbents include clay, cholestyramine, esterified glucomannan, activated charcoal, and other modified polymers, and they absorb OTA, AFB1, ZEN, DON, FB1, and T-2 toxin in the range of 17% to 100% in liquid environments. The use of adsorbents is one of the most cost-effective approaches to remove mycotoxins from food. Nevertheless, the release of toxic materials from chemical adsorbents to food due to the prolonged contact of chemical adsorbents with food, the unwanted reactions between chemical adsorbents and the ingredients, and the removal of adsorbent mycotoxin complexes from foods are the major safety concerns with chemical adsorbents. In addition, the overall sensorial quality and final quality parameters (color, clarity, brix, and titratable acidity) can be adversely affected by chemical adsorbents. The European Union has prohibited some chemical adsorbents for use as decontamination materials in the food industry [2]. In order to prevent the harmful effects of mycotoxins, different parameters are used to determine the effectiveness of each binding additive [164], as shown in Table 4:

#### 5.2.3. Control by Food Additives

Different food-grade additives including citric acid, vinegar, baking powder, and sodium bicarbonate can be used to reduce the mycotoxin concentration in food and beverages. Among these food-grade additives, significantly greater reduction in PAT (from 94.11 to 7.55 µg/L) in apple juice was achieved by sodium bicarbonate, and similar effects were observed following UV irradiation for 30 min [142]. Sodium bicarbonate could be a suitable alternate to UV irradiation for the reduction in the PAT concentration in apple juice, because irradiation requires the use of a sophisticated UV irradiation plant with high energy consumption. The quality parameters, including color, the soluble solids content, and the pH of apple juice can be affected by sodium bicarbonate. However, the odor and color of sodium-bicarbonate-treated juice can be recovered by the addition of citric acid.

### 5.3. Biological Control Methods

Conventional physical and chemical approaches to controlling mycotoxins in food and beverages have been used to date. However, the growing concern regarding the adverse effects on health and the environment related to these traditional physical and chemical methods call for new alternatives. Recent developments in biotechnology have revolutionized our ideas about living organisms and introduced more efficient biological approaches. Advances in biotechnology have brought more effective, novel, healthy, and environmentally friendly tools for the reduction in the mycotoxin content in food, including the use of antifungal biomolecules, microbiological control, and the combination of antifungal biomolecules with microorganisms [34]. However, the release of microbial metabolites in food as well as the absorption of nutrients by microorganisms need to be considered during biological control [170].

#### 5.3.1. Microbiological Control

Recently, many microorganisms, including bacteria, yeasts, and molds, have demonstrated the capability to destroy mycotoxins and are safely used to reduce mycotoxins in food (Table 5). *Lactobacillus plantarum* has been shown to degrade 80% of the PAT concentration during incubation at 37 °C for 4 h with 1 × 10^10^ cells/mL [171,172]. Cell-free supernatants of *L. casei* strains showed inhibitory activity on the growth of *Penicillium* spp. and the generation of toxins (patulin and citrinin) [173]. Karami, et al. [174] screened eight *Lactobacillus* strains from local dairies and 70% of isolates presented with antifungal activity on *Penicillium notatum* and *Aspergillus fulvous**. Gluconobacter oxydans* degraded more than 96% of the PAT after twelve hours of treatment by changing its chemical structure. The genus *Gluconobacter* includes five different species that are safe for human health and are frequently used in food production. Apple juice was inoculated with this bacteria and found to be drinkable after three days of incubation [175].

Commonly used food raw materials for fermentation cover a wide range of foods including cereals, legumes, dairy, and fruits [176], which can be frequently affected by mycotoxins. Consequently, mycotoxins are normally found in the fermentation process. A recent study on the microbial treatment of fungal toxins reported that yeast fermentation entirely degrades some mycotoxins. According to a previous study this method is more promising than other control methods [95]. PAT contamination of fruit-based products and beverages is a serious food safety problem because of the high level of consumption of these products worldwide [177]. Extensive study has been conducted to reveal the impact of yeasts on PAT biodegradation since the 1990s. For example, nearly 90% of PAT was degraded in the reaction medium after three days of yeast fermentation [178]. In a recent study, it was found that the PAT content in fermented apple juice contaminated with yeast during alcoholic fermentation was decreased [179]. The researchers observed that out of eight yeast strains tested, six strains reduced the PAT concentration to below the detection level, while all eight strains showed a reduction in the PAT concentration of 99% or better. Meanwhile, the control stored for the same duration (2 weeks) had only a 10% reduction. The production of PAT-degrading enzymes is induced by the presence of PAT [180]. 

Recently, some researchers stated that *Saccharomyces cerevisiae* converted PAT into ascladiol. It was also reported that the toxicity of ascladiol is only one-fourth that of PAT [181]. However, the biological control of mycotoxins with yeast is limited to products that can be fermented. In addition, yeasts are sensitive to PAT, and the growth of yeast can be completely inhibited when the concentration of PAT is more than 200 µg/mL [182]. There is a growing interest in the use of microbial approaches for the degradation of PAT due to the autonomous reproduction of microbes. Several yeast species such as *Saccharomyces cerevisiae*, *Rhodosporidium paludigenum*, *Rhodosporidium fluviale*, and *Wickerhamomyces anomalus* have biocontrol activity on pathogenic fungi *(Aspergillus japonicas, Aspergillus uvarum, Cladosporium cladosporioides, Aspergillus aculeatus Talaromyces rugulosus, Penicillium georgiense and Penicillium expansum*) and are mainly responsible for postharvest decay in table grapes [37].

#### 5.3.2. Antifungal Biomolecules

Naturally occurring various molecules from organisms can be applied to inhibit PAT-generating molds. It was observed that PAT-producing molds can be prevented by flavanones and their glucoside esters, and 95% of PAT can be reduced in fruits [189]. Deleterious effects of *Penicillium expansum* on the quality of fruits have been reduced by chitosan [190]. The efficacy of antifungal biomolecules on chemical compounds depends on the concentration applied.

#### 5.3.3. Enzymatic Control

The application of enzymes is a promising technique due to its specificity and rapid degradation of mycotoxins in beverages. Over the last few years, there has been growing interest in the enzymatic degradation of mycotoxins because of their detoxification efficiency and safety [191]. Several techniques have been materialized, for example, porcine pancreatic lipase (PPL) immobilized with calcium carbonate, which destroyed 99% of PAT at pH 5.0 and 30 °C over a period of 3 h [192,193]. The best conditions for PPL applied at a concentration of 3 × 10^4^ µg/L were 40 °C and 18 h. After enzymatic treatment, there were no significant changes in the nutritional and sensorial properties of apple juice. Researchers have discovered a novel multi-functional recombinant fusion enzyme named ZHDCP that can degrade OTA, which is a commonly occurring mycotoxin in apple and grape products [194]. Beer is commonly contaminated with DON. Proteases, glucanases, amylases, and other enzymes are used to detoxify DON in beer production [195]. Fruit juices and purees may contain PAT. Enzymatic degradation of PAT has been associated with different species of bacteria and yeast [196]. The Fum8p enzyme produced by *Aspergillus welwitschiae* has the ability to detoxify fumonisins and produce less toxic compounds (FPy and Fla), including those produced by *Fusarium* spp. [197]. However, the cost of the operation might be increased due to the continuous supply of enzymatic materials. Therefore, it is essential to recover the enzymes after use by developing a suitable technique, and identifying new microorganisms for enzyme production is also important [192,193].

#### 5.3.4. Adsorption by Biological Materials

Some microorganisms possess the ability to remove mycotoxins from the food system by adsorbing in their cell walls. An adsorption capability has been found in most gram-positive bacteria and yeasts. The outcomes of recent investigations on mycotoxins adsorption by biological materials are summarized in Table 6. It has been found that about 20% to 90% of mycotoxins could be adsorbed by microorganisms in different liquid food matrices. In comparison with living microorganisms, inactivated microorganisms (heat-treated microorganisms) have demonstrated similar or even better adsorption of mycotoxin in aqueous solutions [198]. Mycotoxin adsorption mechanisms involve physical interactions with the cell walls of microorganisms rather than biological degradation. Thus, mycotoxins do not involve any chemical reactions with the binder during adsorption. Peptidoglycans have been isolated through the purification of cell walls of lactic acid bacteria and showed a higher mycotoxin binding capacity than cell pellets [199,200]. Thus, peptidoglycans play vital functions during the adsorption of mycotoxins and the adsorption efficacy and the number of adsorption sites can be increased by chemical methods such as acid treatment [200]. Recently, animal polysaccharides have attracted more attention from researchers. Assaf, et al. [201] observed that chitin from shrimp shells showed the ability to bind to 17% to 54% of AFM1, depending on the incubation time and concentration of both toxins and chitin. Higher concentration of chitin and longer incubation periods increase the efficiency of adsorption. More than 90% of the AFM1 adsorption can be achieved within 24 h by 0.25 g/mL of unground shrimp shells or 0.15 g/mL of ground shrimp shells. At the same concentration and incubation period, extracted chitin showed higher adsorption rates than both intact and ground shrimp shells. Bioadsorption of PAT was also investigated with superior magnetic chitosan in kiwi fruit juice. Chitosan-coated Fe_3_O_4_ particles have been designed as magnetic adsorbents and were made at a 1:1 ratio of Fe_3_O_4_ particles to chitosan. The formulated magnetic chitosan adsorbed 89% of PAT, and the recovery rate of the adsorbent was nearly 100% after 60 min [202].

Biological materials such as microbial cell walls, peptidoglycans, chitosan, chitin, and enzymes are preferred for food due to their health and environmental friendliness. However, biological materials are less effective than chemical absorbents, and physical and chemical methods are comparatively cheaper than biological control methods. Hence, it is necessary to increase the mycotoxin adsorption capacity of biological adsorbents by designing suitable combinations of physical (high temperature) and chemical (acid-base) treatments that are compatible with the biological adsorbents [203].

## 6. Critical Challenges of Mycotoxins in Beverages

Mycotoxins possess very stable chemical structures that remain unchanged after pasteurization treatment. It has been reported that proper selection, adequate cleaning and washing, and careful sorting of fruits are very crucial factors for the mitigation of mycotoxin contamination during the manufacturing of beverages [210]. As children drink more juices than wine as compared to adults, therefore, the incidence of mycotoxins in fruit juices is a matter of serious concern [211,212].

The detection of mycotoxins in fruit juice and beverages is very challenging. For instance, during the chromatographic analysis of PAT in apple juice, some other compounds showed similar retention under UV detection at a wavelength of 277 nm [213]. At 276 nm 5-hydroxymethylfurfural (5-HMF), pectin, and some proteins showed maximum absorption, which is similar to PAT absorption. This phenomena make difficult for identification via UV or FLD detection systems [214,215]. Several analytical procedures have been established for the detection and quantification of each group of mycotoxins in different food products.

Physical methods can be applied at large and small scales for a wider range of food, but some physical methods including irradiation have negative effects on the nutritional, antioxidant, and sensorial properties of food. Chemical methods are easy to use and comparatively cheap, but their main limitation is the toxicity of residues and secondary products. Additionally, the toxicity of the mycotoxin-degraded products needs to be measured. Although the adsorption of mycotoxins by chemical adsorbents is one of the most inexpensive detoxification methods, the safety of absorbent materials and the removal of the adsorbent–mycotoxin complex from foods is still challenging. In addition, the overall sensorial quality and final quality parameters (color, clarity, brix, titratable acidity, pH, and TSS) can be adversely affected by chemical treatments. Biological control methods are healthy and environmental friendly. However, microbial approaches may deteriorate the food quality by absorbing nutrients and releasing metabolites into the food matrices. Additionally, biological control methods are more expensive than physical and chemical control measures. Another critical challenge is the commercialization of biological control methods by overcoming the limitations in translation from laboratory trials to commercial applications.

## 7. Conclusions

This review summarized the major mycotoxins in fruit juices and beverages and the methods used for their analysis and prevention. Fruit- and cereal-based beverages (fruit juices, wines, beer, and cider) are primarily contaminated by AFs, OTA, PAT, DON, ZEN, T-2, HT-2, and Alternaria toxins which, in small amounts, can be detrimental for humans. The incidence of mycotoxins in food matrices is considered a serious issue. The development of novel and advanced sensor-based early-warning tools for mycotoxin detection would be useful for reducing the risk. The inhibition of fungal growth and subsequent mycotoxin production in raw food materials is preferable for the prevention of mycotoxin contamination in beverages. Sometimes, mycotoxin contamination in beverages cannot be avoided, in spite of taking preventive measures to control mycotoxins in the field and during storage. If mycotoxins are detected in beverages, the use of decontamination and detoxification strategies during postharvest treatment can minimize human exposure to mycotoxins. Different biological, chemical, and physical techniques have been established to limit mycotoxin contamination in beverages. However, so far, no single method has been shown to be 100% effective or applicable for all foods. To achieve the effective degradation and/or detoxification of mycotoxins, more research is required to develop an integrated management strategy by combining multiple control methods.

## Figures and Tables

**Figure 1 toxins-13-00323-f001:**
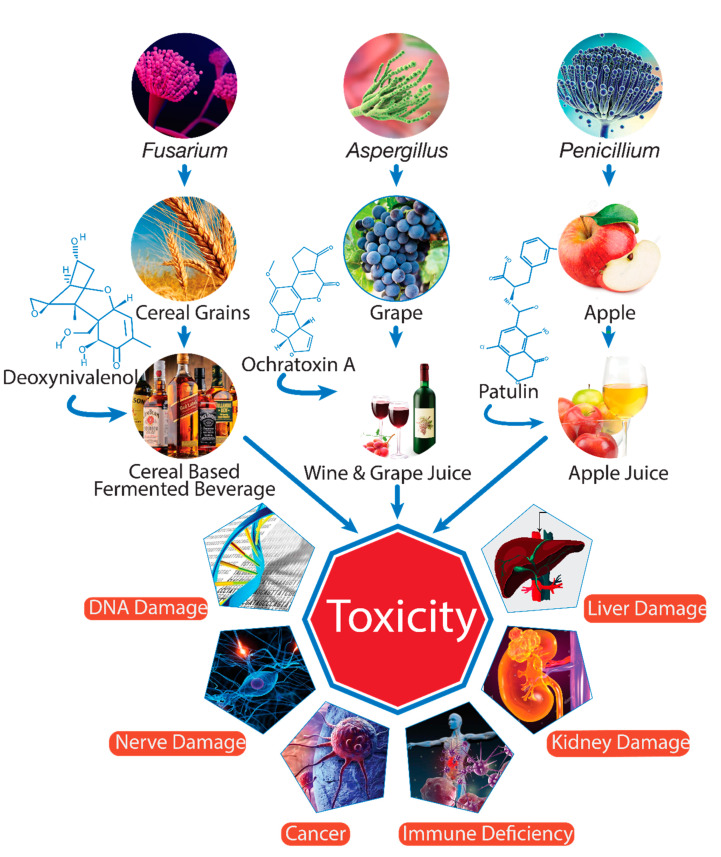
Mycotoxin contamination of beverages and adverse effects on health (drawn using Adobe Illustrator CC software).

**Figure 2 toxins-13-00323-f002:**
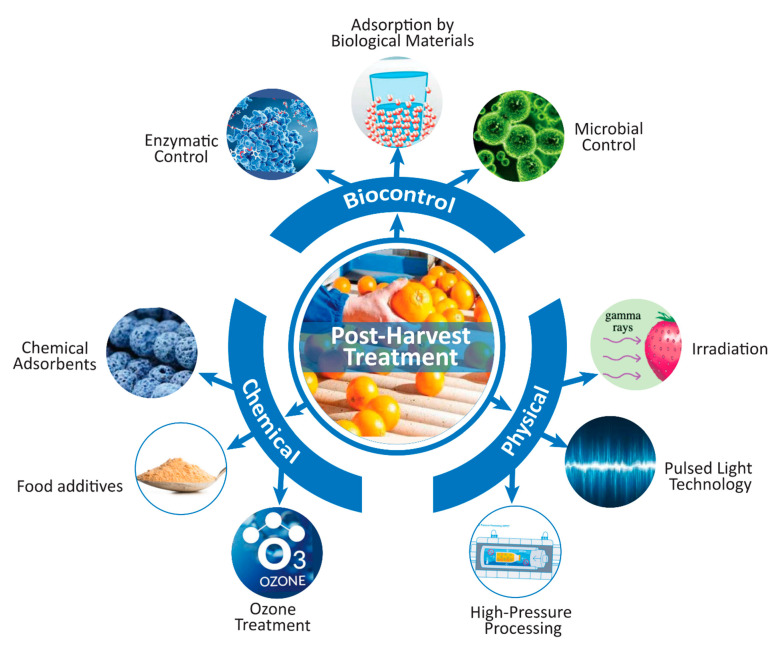
Scheme for reducing the mycotoxin concentration in beverages using postharvest treatments (drawn using Adobe Illustrator CC software).

**Table 1 toxins-13-00323-t001:** Major mycotoxins involved in the contamination of beverages.

Mycotoxins	Products Contaminated	Producing Microorganisms	References
Aflatoxins B1, B2, G1, G2	Orange, apple juice, grape juice, grapefruit peel	*Aspergillus chevallieri*, *A. flavus*, *A. niger*, *A. oryzae*, *A. parasiticus*, *A. repens*, *A. ruber*, *A. tamarii*, and *A. wentii*	[19,20]
Ochratoxin A (OTA)	Grape juice, coffee, beer, and wine	*A. ochraceus*, *A. carbonarius*, *A. niger*, *A. tubingensis*, and *Penicillium expansum*	[19,21]
Patulin (PAT)	Fruit juices	*Penicillium expansum*, *P. patulum*, *Aspergillus clavatus*, *Byssochlamys fulva*, and *B. nivea*	[22,23]
Fumonisins (FBs)	Beer	*Fusarium**proliferatum*, *F. verticillioides*, and *F. nygamai*	[24,25,26]
Trichothecenes (type B: Deoxynivalenol (DON))	Plant-based beverages, beer	*F. graminearum*, *F. cerealis*, and *F. culmorum*	[25,27,28,29]
Trichothecenes (type A: HT-2)	Functional vegetable milks, beer	*F. sporotrichioides*,and *F. langsethiae*	[29,30]
Trichothecenes (type A: T-2 toxin)	Plant-based milks, beer	*F. sporotrichioides*, and *F. langsethiae*	[28,30]
Zearalenone (ZEN)	Beer, wine	*F. graminearum*, *F. culmorum*, *F. equiseti*, *F. cerealis*, *F. verticillioides*, and *F. incarnatum*	[25,31]
Alternaria toxins (TeA, AOH, AME)	Fruit juices, wine, beer	*Alternaria alternate*, *A. tenuissima*, *and A. arborescens*	[32,33]

**Table 2 toxins-13-00323-t002:** Maximum levels for mycotoxins in fruits and their processed products set by the EU [67].

Commodities	MLs (μg/kg)	
**Aflatoxins**	B_1_	B1 + B2 + G1 +G2
Dried fruits subjected to sorting or other physical treatment prior to human consumption or use as raw materials for food production	5.0	10.0
Dried fruits and finished products for direct human consumption or use as raw materials for food production	2.0	4.0
Cereal-based processed products and baby foods for young children and infants	0.10	–
**Ochratoxin A**		
Dried vine fruits (raisins, currants, and sultanas)		10.0
Wines (including sparkling wines and excluding liqueur wines and wines with an alcoholic strength of not less than 15% by vol) and fruit wine		2.0
Aromatised wines, aromatized wine-based cocktails, and aromatized wine-based drinks		2.0
Grape juice, grape nectar, grape must, and reconstituted concentrated grape must and reconstituted concentrated grape juice intended for direct human consumption		2.0
Cereal-based processed products and baby foods for young children and infants		0.50
Instant coffee (soluble coffee)		10
Roasted coffee beans and ground roasted coffee, excluding soluble coffee		5
**Patulin**		
Fruit juices, fruit nectars, and reconstituted concentrated fruit juices		50.0
Ciders, spirit drinks, and other fermented beverages made from apples or apple juice		50.0
Solid apple foodstuffs including apple puree and apple compote for direct consumption		25.0
Solid apple foodstuffs and apple juice, including apple puree and apple compote branded and promoted for young children and infants		10.0
Baby foods other than cereal-based, processed foodstuffs for young children and infants		10.0

**Table 3 toxins-13-00323-t003:** Overview of common detection methods for mycotoxins in beverages as well as beverage-producing crops.

Analytical Methods	Detection Method	Toxin	Applicability	LOD	References	Advantages	Disadvantages
TLC	CCD	Patulin	Apple Juice	14 µg/L	[77]	Time saving, specific fluorescence spot on UV light	Limited plate length and environmental effects on measurement
HPLC	FD	OTA	Wheat	23 pg	[78]	Fast, high resolution data, accurate and easily reproducible. Less training required	Expensive and method development could be challenging
	MS/MS		Wine	0.005 ng/ml	[79]		
	FD			0.09 µg/L	[80]		
		AFs	Food items	1.6-5.2 mg/kg	[81]		
	UV and FD		Milk	0.13–0.16 mg/L	[82]		
LC	FD	OTA	Wine	0.07 ng/ml	[83]	Several mycotoxin detections, high sensitivity, provides confirmation	Expensive, required expertise In case of MS, sensitivity depends on ionization
		ZEN	Barley, Maize, Wheat	100 µg/Kg	[84]		
		AFB1	Corn	2–5 ng/g	[85]		
	MS/MS	Trichothecenes	Wheat and maize	0.2–3.3 µg/Kg	[86]		
Automated microarray chip reader	Chemiluminescence	OTA	Coffee	0.3 µg/L	[87]	High throughput, multiplexed, parallel processing method	Not so common to their variability and reproducibility, require intensive labor
Electro-polymerization onto surface	SPR	ZEN	Corn	0.3 ng/ml	[88]	Suitable for cereals sample, sim- plicity, portability, and ease to use, can be used in field	Optimization and validation not reported for this method
Immunochromatographic strip	Highly Luminescent Quantum Dot Beads	AFB1	Maize	0.42 pg/ml	[89]	A simple method for rapid screening, superior performance	Required expertise
Direct, competitive magneto-immunoassay	SPR	OTA	Beverages	0.042 µg/L	[90]	Rapid, cost effective, and sensitive	
	Electrochemical	FB	Maize	0.33 µg/L	[91,92]		
Lateral flow immunoassay	Colorimetric			199 µg/Kg	[93]	Fast, one-step assay, no washing step, low cost and simple	Qualitative or semi quantitative results, sample volume governs precision
Photonics immobilization technique	Quartz-crystal microbalance (QCM)	Patulin	Apple puree	56 ng/ml	[94]	Specific, higher sensitivity, generality, response (only requires a few minutes), flexibility, and portability	The decrease of the signal in the presence of high analyte concentrations, in situ analysis
Surface-enhanced Raman scattering (SERS)-based immunoassay	Silica-encapsulated hollow gold nanoparticles	AFB1	Food	0.1 ng/ml	[95]	Enhanced ELISA method	Hard to synthesize and expensive
ELISA	UV absorbance	AFM1	Milk	4–25 ng/L	[96]	Fast, simple, economical, high sensitivity, simultaneous analysis of multiple samples, easy to screen	Lack of precision at low concentrations, matrice interference problems, possible false-positive/negative results
		ZEN	Maize	0.02 µg/L	[97]		
		AFB1 and AFM1	Food items	0.13-0.16 mg/L	[82]		
	Electrochemical	FB	Corn	5 µg/L	[98]		

Thin layer chromatography = TLC, High performance liquid chromatography = HPLC, Liquid chromatography = LC, Enzyme-linked immunosorbent assay = ELISA, FD = Fluorescence detection, Ultraviolet = UV, Charge-coupled device = CCD, Surface plasmon resonance = SPR, Mass spectrometry = MS.

**Table 4 toxins-13-00323-t004:** Evaluation parameters to determine the effectiveness of the binding agent.

Evaluation Criteria	References
Effectiveness of the active ingredient confirmed by scientific evidence	[7]
High rate of addition	[165]
Stability over a broad pH range	[166]
High ability to adsorb higher mycotoxin concentrations	[164]
Strong affinity to adsorb low mycotoxin concentrations	[164]
Assertion of chemical interaction of mycotoxin with adsorbent	[167]
Established in vivo data with all relevant mycotoxins	[168]
Non-toxic, environmentally friendly component	[169]

**Table 5 toxins-13-00323-t005:** Adsorption of mycotoxins by bacteria and fungi in different food matrices.

Mycotoxin	Microorganism (Genus)	Strains	Matrices	Effects	Time	Reference
FB	Bacteria (*Enterococcus*)	*E. faecium* 21605	Apple juice	64% Adsorption	24 h	[183]
PAT	Yeast (*Saccharomyces*)	*S. cerevisiae* strain YS3 (laboratory prepared)	Apple juice	70% Adsorption	24 h	[182]
PAT	Yeast (*Saccharomyces*)	*S. cerevisiae* strain YS3 (commercial)	Apple juice	76% Adsorption	24 h	[37]
PAT	Yeast (*Saccharomyces*)	*S. cerevisiae* YS1–YS10	Apple juice	50% to 7% Adsorption	24 h	[184,185]
PAT	Yeast (*Saccharomyces*)	*S. cerevisiae* YS3	Apple juice	100% Adsorption	36 h	[186]
PAT	Yeast (*Saccharomyces*)	*S. cerevisiae*	Apple juice	90% to 96% Adsorption	143 h	[186]
OTA	Yeast (*Saccharomyces*)	*S. cerevisiae* Malaga LOCK 0173	Grape/blackcurrant juice	85% Adsorption	10 days	[187]
OTA	Yeast (*Saccharomyces*)	*S. cerevisiae* Syrena LOCK 0201	Grape/black currant juice	83% Adsorption	10 days	[187]
OTA	Yeast (*Saccharomyces*)	*S. cerevisiae* bakery BS strain	Grape/blackcurrant juice	64% Adsorption	10 days	[187]
OTA	Yeast (*Saccharomyces*)	*S. cerevisiae*	White wine	76% Adsorption	90 days	[188]
OTA	Yeast (*Saccharomyces*)	*S. cerevisiae*	Red wine	86% Adsorption	90 days	[188]
OTA	Yeast (*Saccharomyces*)	*S. cerevisiae*	Rose wine	90% Adsorption	90 days	[188]

**Table 6 toxins-13-00323-t006:** Adsorption of Patulin (PAT) by biological materials in different food matrices.

Bioadsorbents	Adsorption Capacity	Time (h)	Matrices	Reference
Zirconium-based absorbent (UiO-66(NH_2_)	4.4 µg/mg	3	Apple juice	[204]
Nano-Fe3O_4_ modified inactivated yeast	8.6 × 10^−3^ µg/mg	3.5	Apple juice	[205]
Cross-linked xanthated chitosan resin (CXCR)	130.0 µg/mg	18	Apple juice	[206]
Inactivated microbial cells on magnetic Fe_3_O_4_@CTS nanoparticles	90.0%	24	Orange juice	[207]
Superior magnetic chitosan	19.4 × 10^−3^ µg/mg	9	Kiwi juice	[202]
Caustic treated waste cider yeast biomass	58.3%	24	Apple juice	[208]
Non-Cytotoxic Heat-Inactivated Cells and Spores of *Alicyclobacillus* Strains	12.6 × 10^−3^ µg/mg	24	Apple juice	[209]
Inactivated *Lactobacillus rhamnosus* powder	53.39%	4	Apple juice	[203]

## Data Availability

Not Applicable.

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
