# Peer review of "Critical Assessment of Mycotoxins in Beverages and Their Control Measures"

_toxins, 2021, doi:10.3390/toxins13050323_

Round 1

Reviewer 1 Report

The article “Critical assessment of mycotoxins in beverages and its control measures” aims at reviewing the major mycotoxins presence in plant-based beverages, their current detection and measurements methods, and the approaches to their mitigations.   In general, I find the manuscript is well written with a great amount of information, it is well organized and comprehensively described. However, I think the work can be improved and completed by adding two sections:

  1. In “Major mycotoxin in beverages” (chapter 2, line 69): I think is important to add a section about “T-2,HT-2 toxins”; In fact, T-2 and HT-2 toxins are two of the most toxic members of type-A trichothecenes, produced by Fusarium species, especially F.langsethiae and F.sporotrichioides. it is possible to find these mycotoxins in vegetable milks and beer. For the same reasons these mycotoxins should be included in Table 1.
  2. In “Detection and quantification of mycotoxins in beverages” (chapter 4, line 193): I think is important to add a section about “ELISA detection”; In fact there are different ELISA methods to detect DON, T-2,HT-2, ZEA, aflatoxins, etc.. these methods are frequently used because they are rapid and reliable.

Other minor comments:

In References: Please, cited journals should be abbreviated according to ISO 4 rules, see the ISSN Center's List of Title Word Abbreviations or CAS's Core Journals List, as the authors instructions in “MDPI Reference List and Citations Style Guide”

Line 27: please, add space between “species” and “[1]”

Figure 1: please correct "Pinicillium"

Line 133: please, write “The fumonisin B occurs”

Line 134 and line 217: please write the numbers of B1, B2, B3 in the same way.

Line 148: please, add space between “fever” and “[46]”

Line 181: please, add space between “b.w.” and “[61,62]”

Line 411: please, write Fusarium in italics and add space between “Fusarium” and “[125]”

Line 487: please, write “Saccharomyces cerevisiae”

Line 494, line 537: please, write “food matrices”

Line 563: please, don’t write “microbial” with capital letter

Line 571-572: please, add T2-HT2 in the mycotoxins list

In Table 5 (last line): please, write “Lactobacillus rhamnosus

Author Response

Dear Sir/Madam,

Thank you for your time and cooperation for reviewing this manuscript. You have reviewed our manuscript critically. We believe, your effort will definitely improve the quality of our manuscript. We have carefully addressed all the comments. Please find our responses in pdf file.

On behalf of Authors,

Yu Dianzen

23.02.21

Reviewer 2 Report

Toxins Mycotoxins comments:

This review is supposed to be a current overview of the detection and mitigation strategies for mycotoxins.  The authors have done a pretty decent job of assessing these factors in the review but improvements should be made to improve the readability and flow of the review.  Please see below for some suggestions:

  1. A) English language: overall the review is pretty good but word choice/usage and transitions should be looked over with a fine tooth comb to make the review better. Be consistent with singular vs plural usage, i.e. beverage vs beverages, is vs are, etc. Be consistent with scientific names throughout the paper to make sure they are italicized.

  1. B) Figures and Tables… when denoting them in the text should not be Bolded.

  1. C) Please make sure references are correctly formatted in the reference section i.e. journal of agricultural food et.. should they be capitalized?

  1. D) Line 29: It can pose… range from acute.. Please change to it can pose negative health effects ranging from …

  1. E) Line 31/34.. sounds awkward.. what about the most commonly identified mycotoxins in fruit… are comprised of aflatoxins… that are mainly produced by different species of.

F)Line 46-48, As a secondary.. should be As secondary metabolites, mycotoxins are … processed products such as beverages which can pose …

  1. G) Figure 1 title- suggestion: Mycotoxin contamination of beverages and its adverse effects on health

  1. H) When using species ie. Aspergillus. you can write it as Aspergillus spp. and all related identifications.

  1. I) Line 74-85, this paragraph is a bit abrupt in the transitions and sounds awkward. What about- The four major naturally occurring aflatoxins are AFB1 (classified as a Group I carcinogen by the IARC), AFB2, etc.. Please keep in mind that in case of chronic aflatoxicosis, it should be In the case of afltoxicosis. Line 81-82- moreover, the … AFs is intensively connected to cognitive.. what about AFs is significantly correlated with cognitive impairment and growth inhibition in children?  EFSA has set.. should be The EFSA has set- line 84/85.

J)Table 1.  Please change title- maybe Major mycotoxins involved in contamination of beverages. Please be consistent as to when words are capitalized in this Table 1 i.e. Oranges, Grapefruit juice, grapefruit peel –(why not capitalize here?). Also be consistent in this table of abbreviating the scientific names  and not do some one way or some another way.

  1. K) Line 131 Fumonisins (FBs) mycotoxins.. shouldn’t this be Fumonisin (FB) mycotoxins?

  1. L) Section 3.5 DON- too long of sentences that run on and are fragmented. Please rephrase this paragraph.

  1. M) b.w./day—is it supposed to be b.w./day or BW/day or bw/day?

  1. N) Be consistent of writing The EFCSA or The US FDA instead of starting with EFCSA US FDA throughout this paper as appropriate.

  1. O) All of section 4- please rewrite to be coherent. This section is very choppy and doesn’t transition well within each subsection. Line 194—I think you mean mycotoxin levels in contaminated food and beverage can be very low and this necessitates the development of suitable, rapid, and sensitive detection method.  In all the sections, can you please put the limits of detection for each of those assays (LODs).  Additionally, each of the detection methods and mitigation strategies should have clearly defined advantages and disadvantages stated in their relevant section. For example, skilled personnel required, large and or expensive equipment, length of experiment needed- i.e minutes , hours, vs days?, etc.

  1. P) 4.3 Biosensor and immunosensor- Though these assays are common such as Elisas and the such, have there been no advances using lateral flow assays/dipstick tests? Those types of assays are cheap and inexpensive and easily can be used. The downfall is that like for all immune-based assays would be the dependent on the antibodies available.

  1. Q) Section 4.4 . Though the review by Mann et al is useful, one should also acknowledge the seminal original papers that are discussed here.

  1. R) I think Section 4 will be better served with a Table describing the detection methods used , their applicability, advantages/disadvantages, and Limits of detection so it can be referred in Section 4 otherwise this section is way too verbose and confusing.

  1. S) Postharvest or post-harvest, preharvest vs pre-harvest, pretreatment vs pre-treatment, … please be consistent.

  1. T) Section 5.2.2. Line 410—what enzyme? What reference is this related to?

  1. U) Make sure if you state something, then there should be a corresponding reference(s) if important. This review has several incidences that this occurs-especially in Section 4 and Section 5.

Author Response

Dear Sir/Madam,

Thank you for your time and cooperation for reviewing this manuscript. You have reviewed our manuscript critically. We believe, your effort will definitely improve the quality of our manuscript. We have carefully addressed all the comments. Please find our responses in pdf file.

On behalf of Authors,

23.02.21

Reviewer 3 Report

This MS aims to summarize the occurrence of major mycotoxins in fruit beverages and they also mentioned their toxicity and several remediation methods to tackle this problem. 1) the author should clearly indicate their rationale for writing this review paper, yet many review papers are published. 2) Figure 1: not accurate description of the route of exposure and mechanism of action and the affected part of the body. Aspergillus can grow on almost all and penicillium as well, so its not one spp can grow on apple or grape. Penicillium not pinicillim. The mechanism of toxicity and the affected site of toxicity is misleading and should be totally revised to specify the action for individual aflatoxin. The author should mention the source of all these pictures. Figure 2. Biocontrol usually refers to those used in the field like using non-aflatoxin producing A. flavus. Do you think that the enzymatic process is a chemical of biological method for mycotoxins detoxification? Table 4; is Saccharomyces should be classified as yeast or fungi?  Line 538: critical challenges: what do you mean with strong chemical structure? You mean stable? Line 545-552; the information mentioned here are completely not accurate. Detection of mycotoxin in beverages is much much less complicated than solid matrix and the presence of mycotoxin in here is homogenous. In addition, there is many methods for muti-detection of mycotoxins in a single run. Line 548 (concluding remarks?). line 582 (Nowadays, biological control measures 582 are becoming more popular to control mycotoxins in beverages). Actually, its more research topic studied, however, until now, the EU and FDA approved no method for mycotoxin decontamination. Extensive English editing and language reforming is highly recommended.   

Author Response

(The authors gave the same response as above.)

Round 2

Reviewer 2 Report

Changes have improved the manuscript.  Some minor suggestions before acceptance/publication:

Abstract: line 4- suggest are secondary metabolites of filamentous fungi.

Abstract Line 6-7: rephrase .. to human beings owing to their acute toxicity even at low concentrations.

Line 50: rephrase.. distinguishable progress in society has driven reforms in the world beverage market.

Line 82: rephrase to AFx-contaminated food is generally known as aflatoxicosis...

Section 2.2.: OTs are a group of .. Please put are instead of is in the correct places in this section.

Line 114: Table 1 summarizes the major mycotoxins responsible for the contamination of beverages. 

Table 1: In the column of producing microorganisms, please be consistent with Abbreviating the strains all as A.chevalieri, etc. If you abbreviate on strain, they should all be abbreviated in the column.  

Reviewer 3 Report

thank you for submitting the revised version. However, this MS must be entirely improved. By just looking at the abstract, you can find many serious not correct data such as:

Mycotoxins are secondary metabolites of the filamentous fungi that contaminate food products like, fruits, vegetables, cereals, beverages, and other agricultural commodities. Are vegetables and fruits at high risk of mycotoxin contamination like crops? i dont think so.

"Their occurrence in the food chain especially in beverages can be a serious risk to human beings owing to their properties of creating acute toxicity effects even at a small amount of contamination". Acute toxicity is rarely happened over 60 years of  mycotoxin age (aflatoxin), and they should consumed by large dose to have acute toxic effect. Traces of mycotoxins are more accumulative carcinogenic activity.

"The presence of mycotoxins in beverages is are crucial in some cases due to their higher levels than the limit set by regulations" what do you mean by crucial? is this statement is grammertically correct? is this a big conclusion you drawing here that their occurance in food is risky?

"This review aims to summarize the toxicity of the major mycotoxins that naturally occur in beverages, the systematic assays available for their measurement" what do you naturally occur ...do you think there would be different way for mycotoxin rather than colonization of fungi and producing toxins? what do you mean with systemic assay?

"approaches to their mitigation", is this right sentence?